# The Role of Cadherin 12 (CDH12) in the Peritoneal Fluid among Patients with Endometriosis and Endometriosis-Related Infertility

**DOI:** 10.3390/ijerph191811586

**Published:** 2022-09-14

**Authors:** Ksawery Goławski, Robert Soczewica, Joanna Kacperczyk-Bartnik, Grzegorz Mańka, Mariusz Kiecka, Michał Lipa, Damian Warzecha, Robert Spaczyński, Piotr Piekarski, Beata Banaszewska, Artur Jakimiuk, Tadeusz Issat, Wojciech Rokita, Jakub Młodawski, Maria Szubert, Piotr Sieroszewski, Grzegorz Raba, Kamil Szczupak, Tomasz Kluz, Marek Kluza, Mirosław Wielgoś, Ewa Koc-Żórawska, Marcin Żórawski, Piotr Laudański

**Affiliations:** 11st Department of Obstetrics and Gynecology, Medical University of Warsaw, 02-015 Warsaw, Poland; 2College of Inter-Faculty Individual Studies in Mathematics and Natural Sciences, University of Warsaw, 02-097 Warsaw, Poland; 32nd Department of Obstetrics and Gynecology, Medical University of Warsaw, 00-315 Warsaw, Poland; 4Club 35, Polish Society of Gynecologists and Obstetricians, 53-125 Wrocław, Poland; 5Angelius Provita Hospital, 40-611 Katowice, Poland; 6Division of Infertility and Reproductive Endocrinology, Department of Gynecology, Obstetrics and Gynecological Oncology, Poznan University of Medical Sciences, 60-572 Poznan, Poland; 7Department of Reproductive Health, Insitute of Mother and Child in Warsaw, 01-211 Warsaw, Poland; 8Department of Obstetrics and Gynecology, Central Clinical Hospital of the Ministry of Interior, 02-507 Warsaw, Poland; 9Collegium Medicum, Jan Kochanowski University in Kielce, 25-369 Kielce, Poland; 10Clinic of Obstetrics and Gynecology, Provincial Combined Hospital in Kielce, 25-736 Kielce, Poland; 11Department of Gynecology and Obstetrics, Medical University of Lodz, 90-419 Lodz, Poland; 12Department of Surgical Gynecology and Oncology, Medical University of Lodz, 90-419 Lodz, Poland; 13Department of Fetal Medicine and Gynecology, Medical University of Lodz, 90-419 Lodz, Poland; 14Clinic of Obstetrics and Gynecology in Przemysl, 37-700 Przemysl, Poland; 15Department of Obstetrics and Gynecology, University of Rzeszow, 35-310 Rzeszow, Poland; 16Department of Gynecology, Gynecology Oncology and Obstetrics, Institute of Medical Sciences, Medical College of Rzeszow University, 35-310 Rzeszow, Poland; 172nd Department of Nephrology and Hypertension with Dialysis Unit, Medical University of Bialystok, 15-089 Bialystok, Poland; 18Department of Clinical Medicine, Medical University of Bialystok, 15-089 Bialystok, Poland; 19OVIklinika Infertility Center, 01-377 Warsaw, Poland

**Keywords:** endometriosis, infertility, peritoneal fluid, cadherin 12, type II cadherin, neural cadherin

## Abstract

Cadherin 12 (CDH 12) can play a role in the pathogenesis of endometriosis. The aim of this study was to compare the levels of cadherin 12 in the peritoneal fluid between women with and without endometriosis. This was a multicenter cross-sectional study. Eighty-two patients undergoing laparoscopic procedures were enrolled in the study. Cadherin 12 concentrations were determined using the enzyme-linked immunosorbent assay. The level of statistical significance was set at *p* < 0.05. No differences in cadherin 12 concentrations between patients with and without endometriosis were observed (*p* = 0.4). Subgroup analyses showed that CDH 12 concentrations were significantly higher in patients with infertility or primary infertility and endometriosis in comparison with patients without endometriosis and without infertility or primary infertility (*p* = 0.02) and also higher in patients with stage I or II endometriosis and infertility or primary infertility than in patients without endometriosis and infertility or primary infertility (*p* = 0.03, *p* = 0.048, respectively). In total, CDH 12 levels were significantly higher in patients diagnosed with infertility or primary infertility (*p* = 0.0092, *p* = 0.009, respectively) than in fertile women. Cadherin 12 can possibly play a role in the pathogenesis of infertility, both in women with and without endometriosis.

## 1. Introduction

Endometriosis affects approximately 10% of reproductive-age women worldwide but up to 50% of women diagnosed with infertility [1,2]. It is a common, benign, estrogen-dependent, and inflammatory disorder defined as the presence of uterine endometrial tissue outside of the normal location [3,4]. According to the most popular theory created by Sampson in the 1920s, it is the result of retrograde menstruation [5]. Unfortunately, its diagnosis is still a challenge. Only a histopathologic evaluation of a lesion biopsied during surgery (preferably laparoscopy) can make a definitive diagnosis [6,7]. Such an approach remains the gold standard today. However, due to the invasive character of surgery and, most of all, vague symptoms, which overlap with the symptoms involving gynecological and gastrointestinal processes, we face a strong diagnostic delay varying from 7 to 12 years [8,9,10,11,12]. Although the pathogenesis of endometriosis is, to a great extent, unclear, it is still extensively studied regarding many different pro- and anti-inflammatory factors, such as metalloproteinases, growth factors, cytokines, chemokines, miRNAs, and adhesion molecules, which were shown to be differentially expressed in different clinical samples including the plasma, the eutopic endometrium, and the peritoneal fluid [13,14,15,16,17,18,19]. Cadherin 12 (CDH12) is one of the adhesion molecules, which is a type II cadherin and a member of the neural cadherin (N-cadherin) gene family. It plays an important role in calcium-dependent cell-to-cell adhesion [20]. Endometriosis seems to share some similarities in its pathogenesis with cancer, resulting in the presence of cancer-associated mutations in different types of endometriosis [21]. Additionally, CDH 12 can play a role in neoplastic processes by promoting adhesion, migration, invasion, and angiogenesis [22].

CDH12 is proven to be involved in cancer progression of non-small cell lung cancer, salivary adenoid cystic cancer, bladder cancer, and colorectal cancer [23,24,25,26,27]. In the latter case, it affects the epithelial–mesenchymal transition (EMT), the role of which was also demonstrated in the pathogenesis of endometriosis and infertility [28,29,30,31,32,33]. However, its role in the pathogenesis of endometriosis has not been described yet. There are examples of adhesion molecules involvement such as VCAM-1, Nectin-4, JAM-1, or different types of integrins [34,35,36]. An increased concentration of N-cadherin coexists with higher endometrial cell invasiveness and the ability to migrate in patients with diagnosed endometriosis [37]. Regarding infertility, the abovementioned epithelial–mesenchymal transition (EMT) plays a vital role in the development of endometrial fibrosis, which characterizes intrauterine adhesions (IUs)—a common cause of uterine infertility [38]. Pathologic EMT regulates fertility and epithelial functional integrity and promotes fibrosis and tumorigenesis in the reproductive system [39]. Unfortunately, there is also a lack of publications about CDH 12 involvement in the infertility process despite its well-known participation in the EMT process.

As mentioned before, there are no publications regarding the exact role of CDH12 in the pathogenesis of endometriosis and infertility, nor are there any studies regarding the concentrations of this cadherin in the various biological materials collected from patients diagnosed with endometriosis. Simultaneously, it is common knowledge that the peritoneal fluid is a universally recognized biological material that can be investigated in order to better understand the environment and pathogenesis of endometriosis. Based on the above arguments, we decided to examine cadherin 12 concentration levels in the peritoneal fluid of patients with endometriosis, compared with patients without a visible disease, as the potential pathogenetic justification of CDH12 involvement in this process.

## 2. Materials and Methods

Biological material was collected during a multicenter, cross-sectional study, which was undertaken at 8 Departments of Obstetrics and Gynecology in Poland between 2018 and 2019 (project number: 6/6/4/1/NPZ/2017/1210/13522, financed by the Ministry of Health): Department of Obstetrics and Gynecology, Medical University of Warsaw; Angelius Provita Hospital in Katowice; Department of Gynecology, Division of Infertility and Reproductive Endocrinology, Obstetrics and Gynecological Oncology at Poznan University of Medical Sciences; Department of Obstetrics and Gynecology, Central Clinical Hospital of the Ministry of Interior in Warsaw; Clinic of Obstetrics and Gynecology, Provincial Combined Hospital in Kielce; Department of Surgical Gynecology and Oncology, Medical University of Lodz; Department of Gynecology and Obstetrics, Provincial Hospital in Przemysl; Department of Gynecology, Gynecology Oncology, and Obstetrics, Institute of Medical Sciences, Medical College of Rzeszow University.

The patients underwent laparoscopy during which they were diagnosed according to the revised American Fertility Society Classification of Endometriosis [31] and finally confirmed by histopathologic examination. All the patients completed the World Endometriosis Research Foundation (WERF) clinical questionnaire [40]. As controls, patients without visible endometriosis during laparoscopy were recruited. All the patients underwent surgery because of infertility, pelvic pain, and/or ovarian cysts.

Only patients with suspected idiopathic infertility underwent surgery unless an ovarian cyst was also an indication. Thus, later, we also divided patients with endometriosis into two separate groups due to the similarities in disease activity between the stages in each particular group:(1)Stage I or II endometriosis;(2)Stage III or IV endometriosis.

All the above issues resulted in the inclusion of biological material from 82 patients (44 in the study group and 38 controls).

The exclusion criteria were non-regularly menstruating patients (more than 35 or less than 25 days), age at inclusion below 18 or over 40 years old, patients on any form of hormonal therapy during the last 3 months before laparoscopy, malignant or suspected malignant disease, autoimmunological disease, previous and/or current pelvic inflammatory disease, any prior history of pelvic surgery, uterine fibroids, or polycystic ovaries.

The cycle phase was calculated based on the last menstrual period and the average length of the menstrual cycle. Moreover, the phases of the menstrual cycle in women with and without endometriosis were determined by the histological dating of the eutopic endometrial samples collected simultaneously with pathological lesions.

Written informed consent was collected from all the patients, and the study was approved by the Ethics Committee of the Medical University of Warsaw (KB/223/2017).

Peritoneal fluid was collected via Veress needle aspiration under direct visual inspection at the beginning of the laparoscopy in order to avoid contamination with blood. The procedure was performed at all times in accordance with the Endometriosis Phenome and Biobanking Harmonisation Project standard operating procedures [41]. The material collection did not have any impact on the medical management of the patients and was performed in the manner of the Declaration of Helsinki. The aspirated peritoneal fluid was spun in all the studied centers at 1000× *g* for 10 min at 4 °C. The supernatant was transferred to a fresh 10 mL tube (Sarstedt). The time lapse between peritoneal fluid collection and processing was less than 45 min.

Cadherin 12 concentrations in the peritoneal fluid were determined using an enzyme-linked immunosorbent assay (ELISA) kit (SunRedBio, Shanghai, China) with a minimum detectable concentration of 0.050 ng/mL and the intra- and inter-assay coefficients of variation of less than 10% and less than 12%, respectively. The assay range for Cadherin 12 was 0.08–20 ng/mL. The absorbance at 450 nm was measured using a microplate reader (MULTISCAN GO, Thermo Fisher Scientific, Finland).

Cadherin 12 (CDH12) concentrations between different groups of patients were investigated: patients with vs. without endometriosis, patients with different stages of endometriosis, patients with vs. without infertility, patients with vs. without ovarian cysts, depending on the cycle phase and patients’ age.

Outliers were detected and then excluded using a classic statistical domain based on the interquartile range. The groups were compared by a chi-square test for categorical variables. The Mann–Whitney U test and Student’s *t*-test were performed for continuous variables depending on the distribution of variables after testing for normal distribution using the Shapiro–Wilk test. The level of statistical significance was set at *p* < 0.05. Statistical analysis was performed using Python v3.10 programming language.

## 3. Results

### 3.1. Clinical Characteristics of Patients

The statistical analysis including 82 patients (44 with endometriosis and 38 without endometriosis) revealed that there were no statistically significant differences (*p* = 0.4) in cadherin 12 (CDH12) concentrations between the study group (patients with endometriosis) and the control group (patients without endometriosis). The study and control groups were correctly matched (Table 1).

### 3.2. CDH12 Concentration Differences between Patients Depending on Different Factors

Afterward, we closely examined CDH12 concentrations. We sought to investigate the differences in concentrations between the groups of patients studied based on different factors. We aimed to determine if there were possible statistically significant differences in CDH12 concentrations related to age, infertility, stage of endometriosis, presence of ovarian cysts, day of the cycle, and phase of the cycle (Table 2). There were statistically significant differences in CDH12 concentrations in terms of infertility and primary infertility (*p* = 0.009154; *p* = 0.009866, respectively). The mean concentration in infertile women was 7.75 (ng/mL) in comparison to 4.74 (ng/mL) in women without infertility. Among those women suffering from primary infertility, the mean concentration was even higher, at 8.49 (ng/mL), almost doubling that observed in those patients without this disorder, at 4.74 (ng/mL).

### 3.3. Statistical Significance of CDH12 Concentration Differences Based on Stage of Endometriosis and Infertility Factor

Then, the different groups of patients for the statistical significance analysis of CDH12 concentration differences were selected based on their stage of endometriosis and infertility factor. The analysis results are presented below (Table 3 and Table 4).

### 3.4. Characteristics of CDH12 Concentrations Based on Stage of Endometriosis and Infertility Factor

Finally, we present the results showing the characteristics within the different groups of patients based on their endometriosis stage and infertility factor (Table 5 and Table 6), which were analyzed in this study.

## 4. Discussion

The main finding of our study regarded the significantly higher cadherin 12 concentration levels in the peritoneal fluid of patients with primary infertility. Although there are no specific reports on the potential negative effect of cadherin 12 on the fertility of patients with endometriosis, there can be multiple potential mechanisms in which different cadherins may potentially impact fertility.

In a study by Verma et al., the authors investigated the role of cadherins in endometrial receptivity using mouse models [42]. E-cadherin expression was observed both at the pre-receptive and receptive stages followed by reduced levels in the later stages of implantation, while N-cadherin was detectable only at pre-receptive stages.

It was also shown that a reduction in E-cadherin expression in the blastocyst and endometrial cells results in reduced adhesion to the uterine wall [43]. The upregulation of E-cadherin expression interrupts the epithelial–mesenchymal transition and also causes reduced endometrial receptivity, as observed in a study by Zhao et al. [44]. In another study, Zhou et al. investigated the expression of cadherin 6 in the endometrium during the receptive window of infertile and fertile women [45]. The authors determined that the mid-secretory endometrial samples of patients with primary infertility have reduced cadherin 6 expressions compared with the control group.

Bellati et al. examined E- and K-cadherin expressions in the endometrial samples from women with primary infertility (n = 40), recurrent pregnancy loss (n = 12), and healthy controls (n = 24) [46]. The endometrial expression of E- and K-cadherin was lower in primarily infertile women and in women with recurrent pregnancy loss than in fertile controls.

Similarly, a decreased E-cadherin expression was observed in infertile patients with intramural fibroids (n = 18) compared with fertile women (n = 12) in a study by Makker et al. [47].

In another study by Béliard et al., the authors compared E-cadherin expression in endometriosis and the endometrial samples from women with (n = 18) and without (n = 9) endometriosis [48]. It was observed that the expression of E-cadherin and its receptors in endometriosis and the endometrial samples was similar, and there were no significant differences between the study and control groups.

Ishida et al. examined the characteristics of the endometrium in 66 patients with infertility [49]. It was observed that in the case of chronic endometritis, the incidence of the epithelial–mesenchymal transition detected by the loss of E-cadherin and positive N-cadherin expression was significantly higher than in the control group. The authors concluded that the epithelial–mesenchymal transition could negatively alter the implantation mechanisms.

The association between cadherins and another infertility mechanism was analyzed by Wang et al. [50]. The authors determined cadherin 1 to be one of the molecules controlling oocyte meiosis. An increased cadherin 1 expression was observed in mice with depleted centromere protein T, which is a protein responsible for meiotic resumption. Conversely, the overexpression of centromere protein T resulted in decreased cadherin 1 levels.

In our study, we did not observe any differences in cadherin 12 levels between patients with and without endometrial ovarian cysts. Among cadherins, only E-cadherin (CDH1) and N-cadherin (CDH2) concentrations in the endometrial tissue were investigated; decreased concentrations of E-cadherin and increased concentrations of N-cadherin coexisted with higher endometrial cell invasiveness and the ability to migrate in patients with diagnosed endometriosis [37,51]. Decreased E-cadherin concentrations were also noticed in the peritoneal fluid of patients with endometriosis [52], but there are no more details about this study. Sancakli Usta et al. examined the expression of E-cadherin in the eutopic and ectopic endometrial tissue samples collected from endometriomas (n = 32) and compared them with the endometrial tissue samples from healthy controls (n = 30) [53]. It was observed that women with endometrioma had significantly lower E-cadherin expression in their ectopic tissue than healthy controls. There was no difference in E-cadherin expression between the eutopic and ectopic endometrial samples from the endometrioma subgroup. The authors also observed a negative correlation of E-cadherin expression with the incidence of deep infiltrating endometriosis. There were also fluctuations in E-cadherin expression in the eutopic endometrial tissue during the menstrual cycle, with lower levels during the proliferative phase.

Kang et al. reported that the rs8049282 single nucleotide polymorphism of the E-cadherin gene can be associated with primary infertility in patients with ovarian endometriosis, as it was more prevalent in patients with a history of infertility compared with endometriosis patients with fertility success [54].

The primary aim of our study was to compare cadherin 12 levels in the peritoneal fluid of patients with endometriosis and healthy controls; however, no significant differences were observed using the ELISA method.

We are aware that one of the greatest limitations of our study is its restricted number of observations. With more patients enrolled in the study, we would be able to investigate the issue of infertility more comprehensively, without decreasing its statistical power. We plan to study the possible differences in CDH12 concentrations between those patients diagnosed with primary infertility in comparison with secondary infertility in general but also among patients with and without endometriosis. For now, a total number of nine patients diagnosed with secondary infertility, which was used in this study, does not give such an opportunity.

Furthermore, an investigation of the plasma instead of the peritoneal fluid could be an interesting idea, as the plasma is easier to access and, therefore, presents a greater diagnostic potential.

## 5. Conclusions

Based on our study, cadherin 12 (CDH12) in the peritoneal fluid did not show diagnostic potential for endometriosis. However, it can possibly play a role in the pathogenesis of infertility, both in women with and without endometriosis, resulting in statistically significant differences in CDH12 concentrations in the peritoneal fluid between these two groups, compared with fertile women. This can result in a more precise infertility diagnosis and, therefore, more effective treatment.

However, this hypothesis undoubtedly needs further examination in a larger group of patients. Moreover, testing cadherin 12 levels in the plasma can be an idea that is complementary to our work because of the higher diagnostic potential of this biological material in terms of endometriosis as well as infertility.

## Figures and Tables

**Table 1 ijerph-19-11586-t001:** Study group (patients with endometriosis) and control group (patients without endometriosis) characteristics.

Variable	Patients with Endometriosis (n = 44)	Patients without Endometriosis (n = 38)	*p*
CDH 12 concentration (ng/mL)	4.12	3.24	0.41
	(2.11; 10.28)	(2.08; 7.28)	
	7.22	5.61	
Age	30	31.5	0.61
	(28; 34)	(27.25; 35)	
	31.36	30.763	
Day of cycle	12	10.5	0.08
	(10; 20)	(8; 15.75)	
	14.43	12.21	
Infertility	27	20	0.75
	(61.4)	(52.6)	
Primary infertility	23	15	0.6
	(52.3)	(39.5)	
Secondary infertility	4	5	1.0
	(9.1)	(13.2)	
First phase of cycle	26	27	0.37
	(59)	(71)	

Numerical data are presented as median (interquartile range) mean and categorical as number (%). The *p*-values denote overall significance for the Mann–Whitney U test and Student’s *t*-test depending on the distribution of variables after testing for normal distribution using the Shapiro–Wilk test. For nominal variables independence, chi-square test was used.

**Table 2 ijerph-19-11586-t002:** Statistical significance of CDH12 concentrations depending on different factors.

Factor	CDH12 Concentration *p*-Value	Statistically Significant
Infertility	**0.01**	**YES**
Primary infertility	**0.01**	**YES**
Secondary infertility	1.0	NO
Age	0.17	NO
Stage of endometriosis	0.92	NO
Ovarian cysts	0.93	NO
Day of cycle	0.08	NO
Phase of cycle	0.48	NO

The groups were compared by chi-square test for categorical variables. Mann–Whitney U test and Student’s *t*-test were performed for continuous variables depending on the distribution of variables after testing for normal distribution using Shapiro–Wilk test. Statistically significant values are bolded.

**Table 3 ijerph-19-11586-t003:** Statistical significance of CDH12 concentrations between different groups (including patients with infertility).

Group 1	Group 2	CDH 12 Concentration *p*-Value	Statistically Significant
Patients with endometriosis (study group) and patients with infertility	Patients without endometriosis (control group) and Patients without infertility	**0.02**	**YES**
Stage I or II endometriosis and patients with infertility		**0.03**	**YES**
Patients without endometriosis (control group) and patients with infertility		**0.01**	**YES**

The groups were compared using chi-square test for categorical variables. Mann–Whitney U test and Student’s *t*-test were performed for continuous variables depending on the distribution of variables after testing for normal distribution using Shapiro–Wilk test. Statistically significant values are bolded.

**Table 4 ijerph-19-11586-t004:** Statistical significance of CDH12 concentrations between different groups (including patients with primary infertility).

Group 1	Group 2	CDH 12 Concentration *p*-Value	Statistically Significant
Patients with endometriosis (study group) and patients with primary infertility	Patients without endometriosis (control group) and Patients without primary infertility	**0.02**	**YES**
Stage I or II endometriosis and patients with primary infertility		**0.05**	**YES**
Patients without endometriosis (control group) and patients with primary infertility		**0.03**	**YES**

The groups were compared using chi-square test for categorical variables. Mann–Whitney U test and Student’s *t*-test were performed for continuous variables depending on the distribution of variables after testing for normal distribution using Shapiro–Wilk test. Statistically significant values are bolded.

**Table 5 ijerph-19-11586-t005:** Characteristics of CDH 12 concentrations (ng/mL) within different groups (including patients with infertility).

Group	Median	Mean	Q1	Q3
Patients with endometriosis (study group) and patients with infertility	3.86	8.0	2.58	13.86
Stage I or II endometriosis and patients with infertility	4.29	7.61	3.23	9.88
Patients without endometriosis (control group) and patients with infertility	4.46	7.41	3.04	11.51
Patients without endometriosis (control group) and patients without infertility	2.33	3.61	1.47	3.91

The groups were compared using chi-square test for categorical variables. Mann–Whitney U test and Student’s *t*-test were performed for continuous variables depending on the distribution of variables after testing for normal distribution using Shapiro–Wilk test.

**Table 6 ijerph-19-11586-t006:** Characteristics of CDH 12 concentrations (ng/mL) within different groups (including patients with primary infertility).

Group	Median	Mean	Q1	Q3
Patients with endometriosis (study group) and patients with primary infertility	4.73	8.58	2.58	17.92
Stage I or II endometriosis and patients with primary infertility	4.73	7.93	3.13	10.24
Patients without endometriosis (control group) and patients with primary infertility	5.09	8.34	2.98	12.88
Patients without endometriosis (control group) and patients without primary infertility	2.41	3.83	1.55	4.69

The groups were compared using chi-square test for categorical variables. Mann–Whitney U test and Student’s *t*-test were performed for continuous variables depending on the distribution of variables after testing for normal distribution using Shapiro–Wilk test.

## Data Availability

Not applicable.

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
