# Peer review of "The Role of Cadherin 12 (CDH12) in the Peritoneal Fluid among Patients with Endometriosis and Endometriosis-Related Infertility"

_ijerph, 2022, doi:10.3390/ijerph191811586_

Round 1
Reviewer 1 Report
The authors present a research article concerning the role of Cadherin 12 in the peritoneal fluid among patients with endometriosis. The manuscript is well written in terms of clarity, style, and use of English and has a logical construction. The results presented show that peritoneal fluid CDH12 cannot provide diagnostic help for endometriosis. However, CDH12 concentrations in the peritoneal fluid can possibly play a role in the pathogenesis of infertility, both in women with and without endometriosis. This could assist in infertility diagnosis and then more effective treatment. The discussion section explains the results of the study in the context of published information. The conclusions accurately and clearly explain the main clinical message. The tables are of good quality and helpful in understanding the clinical message. The references are appropriate and current. I can t detect any major flaws in the study. The only concern is that the number of patients involved in the study is small.
Author Response
Dear Reviewer,
Thank you very much for your attention and the review report that you have written. It is really motivating that someone finds our work interesting and accurate.
We are aware that number of patients involved in the study could be bigger. However, at this point we were unable to change this situation. The study was designed with profound attention after thorough analysis. It resulted in multiple inclusion as well as exclusion criteria that had to be met by patients enrolled in the study. Thus, it was a quite big logistic challenge influenced by financial as well as human resources. Nevertheless, we are planning to expand our study group in the future during further investigation of pathogenesis of endometriosis and infertility using different biological materials such as plasma.
Yours sincerely,
Piotr Laudański, Ksawery Goławski & all the authors
Reviewer 2 Report
The article is related to very urgent problem, new tools in early diagnostics of endometriosis. The central molecule of the survey, CDH12, has not previously been investigate and the presented results are absolutely unique. At the same time, authors used peritoneal fluid to investigate CDH12 level in patients with endometriosis. This method could not facilitate the diagnostics of endometriosis but can shed lite into its pathogenesis. So, this investigation has more fundamental than practical value. I suppose that in future the authors should widen this investigation by measuring the CDH 12 level in plasma (as they mentioned in the conclusion) as well as by immunohistochemical evaluation of CDH12 expression in the endometrioid tissue.
The most important result of the investigation is the deference between fertile patients infertile with endometriosis in CDH12 level. At the same time, authors proposed 4 huge tables with a lot of data while most of them are statistically insignificant.
Discussion is written pretty well but I suppose that the data about primary and secondary infertility should be more in focus because the authors revealed the difference between primary infertile and fertile patients with endometriosis and did nor reveal any differenced between secondary infertile and fertile patients with endometriosis (P5L170-173). It should be also mentioned that the majority of cited references are rather old (67% older than 5 years); although some of them are fundamental (like paper by Sampson J. Peritoneal endometriosis due to the menstrual dissemination of endometrial tissue into the peritoneal cavity), others can be replaced by more recent ones. For example, instead of ESHRE guideline 2014 (ref. â„– 6) new ESHRE guideline 2022 should be cited (ESHRE Guideline Endometriosis 2022 (https://www.eshre.eu/Guideline/Endometriosis); instead of Laparoscopic surgery for endometriosis. Cochrane database Syst Rev 2014 (ref. â„–7) a new one “Laparoscopic surgery for endometriosis 2020 Cochrane Database of Systematic Reviews” (https://doi.org/10.1002/14651858.CD011031.pub3) should be cited.
More detailed comments:
P2L66: Although the pathogenesis of endometriosis is to a great extent unclear different it is still extensively studied and many pro- and anti-inflammatory factors: the meaning of this sentence is not pretty clear
P5L169-71: the p value should have equal numbers after the point and not so many (six)
P8L255 (table 4a) and P9L264 (table 4b): the data should not be presented in the tables; it takes too much space and does not enough representative.
P6L213 (table 3a) and P7L241 (table 3b): these tables should be organized clearer because it is not obvious how to compare patients from group 1a and group 1b with patients with group 2a and group 2b. In addition, the tables are excessive and should be reduced by removing statistically unsignificant data.
I recommend to eliminate the mentioned above defects and finalize the text to make it more readable and laconic
Author Response
Dear Reviewer,
We would like to deeply thank you for your comments about our work after thorough and careful reading.
You can find our answers below:
Ad P2L66 - we modified this sentence to be more transparent.
Ad. P5L 169-71 we wanted to be precise from the statistical point of view. However, based on your comment, we changed p values to two numbers after the point.
Ad. P8L255 and P9L264 This is the comment that we would like to explain. We are deeply convinced that presenting the characteristics of patients using tables is the most efficient attitude. Otherwise, the data would be chaotic and extremely hard to be analyzed. In tables there is a scheme of data presentation that is applied for different groups and therefore gives the reader an opportunity to visually encompass all the numbers at one time and find the issues that are interesting just in a few seconds. Nevertheless, we excluded groups that were statistically insignificant in the previous analyses what resulted in smaller and therefore more transparent tables.
Ad. P6L213 and P7L241 We made amendments (by the exclusion of division into 1a) 1b) 2a) and 2b) groups what can be misguiding) so that the tables 3a and 3b will be more obvious regarding which groups are compared with.
Moreover, we have changed references due to your direct suggestions adding also a few references originating more in recent 5 years. However, the others represent theoretical background of our research that was in most of cases not reviewed or changed during recent years. That is the reason why some of the references may look not so up to date.
Taking discussion into consideration, we wanted to focus on general infertility but also primary infertility what was corresponding with our results. Undoubtedly, we would also like to investigate the subject of secondary infertility, but unfortunately number of patients enrolled in our study with secondary infertility (n=9) was too small to present a sufficient statistical power.
Additionally, we have also improved our materials & methods section, adding more details concerning data collection and clinics that were involved in the study.
We hope that you can easily find all the amendments which are in red color.
Yours Sincerely
Piotr Laudański, Ksawery Goławski & all the authors
Reviewer 3 Report
Esteemed Authors and Editorial Team,
In my opinion it seems strange to try to find a link between a cadherin expressed specifically in the brain and neural tissue and endometriosis. It is true that this has never been studied before, however, there is insufficient background in the introduction and in literature to support this choice of biomarker.
Secondly, the title and aim the paper state as interest the possible involvement of CDH12 in endometriosis, wheres the conclusions (correctly drawn from the statistical analysis and results) refer to a completely different matter, that is the role of cadherin 12 in the pathogenesis of infertility.
I think the whole idea behind the study should be reconsidered and I consider the article unfit for publishing in the current form.
Author Response
Dear Reviewer,
Please see the attachment. PDF format will be much more transparent.
Yours sincerely,
Piotr Laudański, Ksawery Goławski & all the authors

Round 2
Reviewer 3 Report
Dear authors,
I appreciate the changes made to the manuscript. I still feel the potential connection between CDH 12 expression and infertility should be addressed in the introduction, aside from having appended the title.
Otherwise the paper looks better in the current format.
Author Response
Dear Reviewer,
Thank you very much again for your comments and big help in improving our article. We have addressed the potential link between CDH 12 and infertility, what you can easily find in the introduction.
All the amendments are painted in orange color.
Yours sincerely,
Piotr Laudański, Ksawery Goławski & all the authors